# LEARNING K-U-NET IN CONSTANT COMPLEXITY WITH APPLICATION TO TIME SERIES FORECASTING

## ABSTRACT

Training deep models for time series forecasting is a critical task with an inherent challenge of time complexity. While current methods generally ensure linear time complexity, our observations on temporal redundancy show that high-level features are learned 98.44% slower than low-level features. To address this issue, we introduce a new exponentially weighted stochastic gradient descent algorithm designed to achieve constant time complexity in deep learning models. We prove that the theoretical complexity of this learning method is constant. Evaluation of this method on Kernel U-Net (K-U-Net) on synthetic datasets shows a significant reduction in complexity while improving the accuracy of the test set.

## 1 INTRODUCTION

The task of training deep models for time series forecasting is pivotal across a broad spectrum of applications, ranging from meteorology to smart city management(Nie et al., 2023). These models play a crucial role in learning complex patterns and making accurate predictions about future events based on historical data. However, a significant challenge that arises in the development of such models is managing the inherent time complexity in training. Traditional approaches generally operate within a framework that ensures linear time complexity. Yet, through meticulous analysis of training on U-Net, we have observed that up to $98\%$ of data involved in low-level training processes can be considered redundant. This redundancy not only strains computational resources but also extends training durations unnecessarily, impacting the efficiency of model development.

To combat this inefficiency, our research introduces a novel algorithm called exponentially weighted stochastic gradient descent with momentum (EW-SGDM), specifically designed to reduce the time complexity from linear to constant. This innovative approach optimally focuses computational power by ignoring redundant data during the learning process, thereby enhancing efficiency. We have empirically demonstrated that the lower bound of this method's time complexity can reach constant levels, showing a potential theoretical limit to how much computational overhead can be reduced while still maintaining robust model training.

We examined its application on Time series forecasting tasks with Kernel U-Net(You et al., 2024). This architecture is engineered to accept custom kernels, by separating the patch manipulation and kernel operation. Therefore providing conveniences for applying complexity reduction on various types of U-Net. The effectiveness of the application of algorithm EW-SGDM on Kernel U-Net was evaluated through a series of tests across diverse time series forecasting datasets. These evaluations have demonstrated that Kernel U-Net achieves a significant reduction in computational complexity while maintaining comparable accuracy levels.

We summarize the contributions of this work as follows:

1. **Introduction of EW-SGDM for Complexity Reduction**: We propose a novel exponentially weighted stochastic gradient descent with momentum (EW-SGDM) algorithm that reduces the time complexity of training deep models from linear to constant.

2. **Integration with Kernel U-Net Architecture**: We adapt EW-SGDM to the Kernel U-Net, an architecture that separates patch manipulation and kernel operations, making it flexible for various U-Net variants.

Figure 1: The Exponentially Weighted Gradient with Momentum (EW-SGDM) Algorithm on U-Shape Architecture. The weight $W = S^l$ is applied on the gradients of the parameters in kernels $\{\phi_1, \ldots, \phi_6\}$ at each level of U-Net, where $S$ is the patch size.

3. **Empirical Validation Across Diverse Time Series Benchmarks**: Extensive experiments across multiple time series forecasting benchmarks demonstrate that the Kernel U-Net with EW-SGDM not only significantly reduces computational complexity but also maintains high accuracy levels, proving its effectiveness.

These contributions collectively advance the field of time series forecasting by addressing key inefficiencies in training U-shape models. Upon completion of the peer review process, we will make the code publicly available.

## 2 METHOD

In this section, we commence by defining the scope and fundamental concepts of time series forecasting, which involves predicting future values based on previously observed temporal data sequences. In the next, we explain the learning complexity and computation complexity with various models. Section 2.3 introduces temporal redundancy on low-level kernels in the U-Net where significant portions of data in a time series may be repetitive and not contribute new information. Section 2.4 presents the algorithm called Exponentially weighted SGD with momentum (EW-SGDM). This method adds weights to the gradients on high-level parameters of the kernel. Section 2.5 analyses the complexity of learning kernels and confirms the constant complexity.

### 2.1 PRELIMINARY

Let us note by $x \in R^{N \times M}$ the matrix which represents the multivariate time series dataset, where the first dimension $N$ represents the sampling time and the second dimension $M$ is the size of the feature. Let $L$ be the length of memory or the look-back window, so $(x_{t+1,1}, ..., x_{t+L,M})$ (or for short, $(x_{t+1}, ..., x_{t+L})$) is a slice of length $L$ of all features. It contains historical information about the system at instant $t$.

In the context of time series forecasting, the dataset is composed of a time series of characteristics $x$ and future series $\hat{x}$. Let $x_t$ be the feature at the time step $t$ and $L$ the length of the look-back window. Given a historical data series $(x_{t+1}, ..., x_{t+L})$ of length $L$, time series forecasting task is to predict the value $(\hat{x}_{t+L+1}, ..., \hat{x}_{t+L+T})$ in the future T time steps. Then we can define the basic time series forecasting problem:

$$(\hat{x}_{t+L+1}, ..., \hat{x}_{t+L+T}) = f(x_{t+1}, ..., x_{t+L}) \tag{1}$$

, where $f$ is the function that predicts the value $(\hat{x}_{t+L+1}, ..., \hat{x}_{t+L+T})$ in the future T time steps based on a series $(x_{t+1}, ..., x_{t+L})$.

| Method | Computation Complexity | Prediction Length | Parameters Update Steps | Learning Complexity |
|--------|------------------------|-------------------|-------------------------|---------------------|
| Linear Matrix | $O(T^2)$ | $O(T)$ | $O(1)$ | $O(T^2)$ |
| ARMA/ARIMA | $O(T)$ | $O(1)$ | $O(1)$ | $O(T)$ |
| Pyraformer | $O(T)$ | $O(1)$ | $O(1)$ | $O(T)$ |
| PatchTST | $O(T^2)$ | $O(T)$ | $O(1)$ | $O(T^2)$ |
| K-U-Net | $O(T)$ | $O(T)$ | $O(1)$ | $O(T)$ |
| EW+K-U-Net | $O(T)$ | $O(T)$ | $O(T)$ | $O(1)$ |

Table 1: Comparison of Computation and Learning Complexity, for Various Methods.

## 2.2 COMPUTATION COMPLEXITY AND LEARNING COMPLEXITY

In this part, we define the computation complexity, the prediction length, the parameter update times, and lastly the learning complexity (Table 1) **??**.

The computation complexity calculates the number of iterations over input or output length $T$. Linear Matrix (NLinear) (Zeng et al., 2023) contains a matrix of size $T^2$ thus its computation complexity is $O(T^2)$. The parameters in ARMA and ARIMA are vectors of length $T$ therefore the computation complexity is $O(T)$. Pyraformer (Liu et al., 2022) processes sequences hierarchically so that the computation complexity is $O(T)$ for the bottom layer. PatchTST (Nie et al., 2023) contains a transformer layer of complexity $O(\frac{T}{S} \frac{log(T)}{S})$ but its linear matrix flatten layer increases the complexity to $O(T^2)$, where $S$ is the patch size.

*Kernel U-Net* (K-U-Net) (You et al., 2024), a unified U-shape architecture that separates the kernel operation and patch manipulation, offering flexibility and improved computational efficiency for time series forecasting. The architecture preserves the essential encoder-decoder structure of U-Net, where the encoder compresses the input time series into latent vectors, and the decoder symmetrically reconstructs the time series. Kernel U-Net guarantees linear complexity $O(T)$ if applying quadratic complexity kernels starting from the second layer.

The prediction length is computed based on the output length. In general, One-step forecasting models such as ARMA, ARIMA, and Pyrafromer predict $O(1)$ length output. Multi-step forecasting models such as Linear Matrix, PatchTST, and K-U-Net predict $O(T)$ length output.

The parameter update steps are the smallest gradient update step in one set of parameters in the model. ARMA, ARIMA, and Linear Matrix update $O(1)$ step gradient after one training epoch. PatchTST, while its transformer layer updates $O(T^2)$ step gradient for $Q$ $K$ matrix, its $V$ matrix updates $O(T)$ step gradient and its flattened layer (also a linear matrix) updates $O(1)$ step gradient. K-U-Net updates $O(T)$ step gradient at low-level kernels and updates $O(1)$ step gradient.

The learning complexity is the division of maximum computation complexity over parameter update steps. Most algorithms in this list keep the same to the computation complexity, while Exponentially Weighted SGD with Moment (EW-SGDM) reweights the highest parameter update steps in K-U-Net by $W = S^{l-1}$, making it possible to update $O(T)$ steps. Thus the learning complexity of EW+K-U-Net becomes $O(1)$ (Section 2.4, 2.5).

## 2.3 TEMPORAL REDUNDANCY

Current literature concentrates on reducing redundant patches by designing strategies over similar patches(Dutson et al., 2023), or reducing frames by heuristic functions(Mathias Parger, 2022). Our observation concentrates more on the temporally overlapping patches in backpropagation. Let us recall that $L$ is the input length of the trajectory segment $x_t$ and $S$ is the patch length. By creating patches from a given time series trajectory $x_t$, we observe nearly $\frac{L}{S}$ times overlapping patches, which offers us spaces for potential complexity reduction (Figures 1). For example, in case $L = 512$ and $S = 8$, the redundancy is $1 - \frac{S}{L} = 98.44\%$

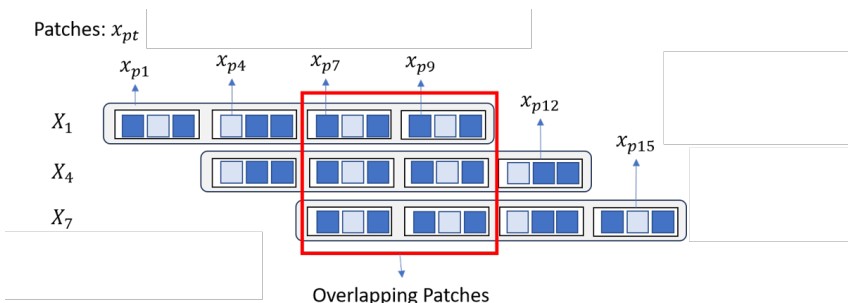

Figure 2: The phenomena of overlapping patches. The low level features are learnt $\frac{L}{S}$ more times by sliding window.

In the context of using Kernel-U-Net, increasing the steps in gradient passing is equivalent to reducing the learning complexity. As kernel U-Net offers meta-operation over patches, which is in general independent of the choice of kernels, this reduction can be simply done within the kernel wrapper. This is a direct advantage of the choice of separating patch operation and kernel manipulation.

## 2.4 EXPONENTIALLY WEIGHTED SGD WITH MOMENT (EW-SGDM)

In this work, we propose a novel approach to address the redundancy and imbalance in the gradient updates across different layers of U-Net architectures. Traditional Stochastic Gradient Descent (SGD) tends to apply uniform updates across layers, which can lead to suboptimal learning dynamics. Layers responsible for capturing low-level features often receive gradient updates of a similar magnitude as those capturing high-level abstract features, despite the difference in the nature of the information being processed at different stages of the network. This uniformity can slow down convergence, particularly in deep networks where the interaction between low- and high-level features is crucial for performance.

Our approach mitigates this issue by introducing an exponentially weighted gradient update mechanism. Specifically, we compute a weight for each layer, denoted as $W^{(l)}$, where $l$ refers to the depth of the layer within the network. This weight $W^{(l)}$ is applied to the gradient update of the corresponding layer, effectively scaling the contribution of gradients for low-level and high-level features differently. The weights are computed as a function of the layer depth, such that:

$$W^{(l)} = S^{l-1} \tag{2}$$

where $S$ is a hyperparameter that controls the rate at which the weights decay (or grow) as we move through the layers. By adjusting the weight $W^{(l)}$, we can prioritize updates to the deeper layers, which capture more abstract, high-level representations, or alternatively, emphasize updates to the shallow layers responsible for low-level details.

This exponentially weighted SGD method adjusts the imbalance in update ratios between low- and high-level features by allowing more fine-grained control over the learning dynamics. The deeper layers, which generally have smaller gradients due to vanishing gradient effects, benefit from larger weight factors $W = L_1^{(l)}$, thereby accelerating their learning. Conversely, the earlier layers, which deal with more fine-grained, local features, can have their updates scaled down, preventing the dominance of lower-level features in the learning process.

The proposed weighting mechanism not only balances the learning across layers but also helps the model converge faster and more efficiently. It reduces the risk of overfitting to specific layers or features, as it ensures that both high-level and low-level representations are adjusted appropriately during training. Empirical evaluations demonstrate that this method results in more stable training and improved performance in various time series forecasting tasks, particularly when applied to deep U-shape architectures like K-U-Net.

---

**Algorithm 1** Exponentially Re-weighted Gradient

---

**Input**: kernel $\phi^{(l)}$, input latents $x^{(l)}$, learning rate $\eta$, layer index $l$, patch size $S$.
**Output**: Updated weights
    # Define the backward function :
    **def** backward($x^{(l)}$, $l$, $S$, $\eta$):
        # Compute the weight
        $W = S^{l-1}$
        # Re-weight the gradient
        $W \nabla_{\phi^{(l)}} \mathcal{L}(x^{(l)}) = S^{l-1} \nabla_{\phi^{(l)}} \mathcal{L}(x^{(l)})$
        # Update the weight with the gradients
        $\theta_{\phi^{(l)}} = \theta_{\phi^{(l)}} - \eta S^{l-1} \nabla_{\phi^{(l)}} \mathcal{L}(x^{(l)})$
        **return** $\theta_{\phi^{(l)}}$

---

## 2.5 Constant Complexity

We give the formal analysis of the constant complexity of proposed algorithm. We assume the dataset is of length $N$ and padded with $0$ to be $N$ couple of input and output pairs.

Given a kernel U-Net, input, and output of length $L, T$, and $L = T$, and $T = S^l$. For a layer $l$, we have a kernel $\phi^{(l)}$ and its exponential weight $W^{(l)} = S^l$.

we define non-redundant update of gradient in kernel $\phi^{(l)}$ by :

$$\mathbb{E}_u[\nabla_{\phi^{(l)}} \mathcal{L}(x)] = \frac{1}{N \cdot S} \sum_{i=1}^{N} \nabla_{\phi^{(l)}} \mathcal{L}(x_i^{(l)}, \ldots, x_{i+S-1}^{(l)}) \tag{3}$$

we define the expectation of total gradient passed at kernel $\phi^{(l)}$ after one epoch:

$$\mathbb{E}[\nabla_{\phi^{(l)}} \mathcal{L}(x)] = \frac{1}{N} \sum_{i=1}^{N} \nabla_{\phi^{(l)}} \mathcal{L}(x_i, \ldots, x_{i+T-1}) \tag{4}$$

$$= \frac{1}{N} \sum_{i=1}^{N} \sum_{j=1}^{T/S^l} \nabla_{\phi^{(l)}} \mathcal{L}(x_{i+(S-1)j}^{(l)}, \ldots, x_{i+(S-1)j+S-1}^{(l)}) \tag{5}$$

$$= \frac{1}{N} \sum_{i=1}^{N} T/S^l \nabla_{\phi^{(l)}} \mathcal{L}(x_i^{(l)}, \ldots, x_{i+S-1}^{(l)}) \tag{6}$$

$$\tag{7}$$

Now we applied the weights to the gradients :

$$\mathbb{E}_{W^{(l)}}[\nabla_{\phi^{(l)}} \mathcal{L}(x)] = \frac{1}{N} \sum_{i=1}^{N} W^{(l)} T/S^l \nabla_{\phi^{(l)}} \mathcal{L}(x_i^{(l)}, \ldots, x_{i+S-1}^{(l)}) \tag{8}$$

$$= \frac{1}{N} \sum_{i=1}^{N} S^{l-1} T/S^l \nabla_{\phi^{(l)}} \mathcal{L}(x_i^{(l)}, \ldots, x_{i+S-1}^{(l)}) \tag{9}$$

$$= \frac{T}{N \cdot S} \sum_{i=1}^{N} \nabla_{\phi^{(l)}} \mathcal{L}(x_i^{(l)}, \ldots, x_{i+S-1}^{(l)}) \tag{10}$$

$$= T \mathbb{E}_u[\nabla_{\phi^{(l)}} \mathcal{L}(x)] \tag{11}$$

now we explain why the exponentially re-weighted algorithm updates $T$ times non-redundant latent vectors in one epoch. Thus the complexity of learning a forecasting task on K-U-Net with a sequence of length $T$ is constant: $O(\frac{T}{T}) = O(1)$.

## 3 RELATED WORKS

**Stochastic Gradient Descent Methods** Optimization plays a critical role in training machine learning models, especially deep neural networks. Various optimization algorithms have been proposed to minimize loss functions efficiently. This section discusses three optimization methods: Stochastic Gradient Descent (SGD), Stochastic Gradient Descent with Momentum (SGDM), and Adaptive Momentum Estimation (Adam).

Stochastic Gradient Descent (SGD) is one of the earliest and most fundamental optimization techniques used in machine learning (Robbins & Monro, 1951). In contrast to the traditional gradient descent, which calculates the gradient of the loss function using the entire dataset, SGD approximates this gradient using a small, randomly selected batch of data samples. This approach not only reduces the computational cost per iteration but also introduces noise, which can help the optimization escape shallow local minima (Bottou, 2010). However, vanilla SGD suffers from slow convergence, particularly in cases where the loss surface is not smooth or contains sharp curvatures.

To address some of these limitations, the introduction of momentum into the gradient updates led to Stochastic Gradient Descent with Momentum (SGDM) (Qian, 1999). Momentum incorporates a moving average of past gradients to smooth out the oscillations that can arise when training with high learning rates. As a result, SGDM accelerates convergence, especially in scenarios with narrow valleys and plateaus in the loss landscape (Sutskever et al., 2013). By adding a momentum term, SGDM not only reduces the variance in the gradient updates but also helps traverse the loss surface more effectively.

Adam (Adaptive Momentum Estimation) further builds on the ideas of SGD by adapting the learning rate for each parameter based on first and second SGDMs of the gradient (Kingma & Ba, 2015). Adam computes individual adaptive learning rates for different parameters, which makes it particularly useful for problems with sparse gradients or nonstationary objectives. The combination of momentum-like behavior and learning rate adaptation has made Adam a popular choice in many deep-learning applications. However, some studies have raised concerns about the generalization properties of Adam compared to SGD in certain settings (Wilson et al., 2017).

Despite the effectiveness of these methods, there remains ongoing research to improve convergence rates, stability, and generalization capabilities. Recent works have explored adaptive variants of SGD and methods that combine the benefits of multiple optimizers (Reddi et al., 2018). Moreover, understanding the theoretical foundations behind the success and limitations of these methods continues to be a critical area of investigation.

### 3.1 EVOLUTION OF U-SHAPE ARCHITECTURES FOR TIME SERIES FORECASTING

Time series forecasting has seen notable advancements in recent years, with deep learning models becoming increasingly effective at capturing temporal dependencies and complex data patterns. Among these, U-shape architectures, which originate from the U-Net model developed for image segmentation (Ronneberger et al., 2015), have gained significant traction. The core of the U-shape design lies in its symmetric encoder-decoder framework. In this framework, the encoder progressively compresses the input data, capturing high-level features, while the decoder restores the data's original resolution, aided by skip connections that link matching layers in the encoder and decoder paths.

The use of U-shape architectures in time series forecasting began with the adaptation of the U-Net structure for one-dimensional signals (Madhusudhanan et al., 2023)Wang et al. (2024). These models are particularly well-suited for capturing both short- and long-term dependencies, which are critical in tasks such as multivariate time series forecasting. A significant advantage of the U-shape design in time series forecasting is its ability to retain intricate temporal details, facilitated by skip connections, which help preserve essential information throughout the down-sampling process (Weninger et al., 2014).

Recent developments have introduced various U-shape architecture enhancements, aiming to improve their efficacy in forecasting. These include hybrid approaches that integrate U-shape networks with attention mechanisms (You et al., 2024) Madhusudhanan et al. (2023), as well as architectures that embed recurrent layers, such as Long Short-Term Memory (LSTM) units (You et al., 2024). These modifications enhance the model's ability to selectively focus on important temporal features while

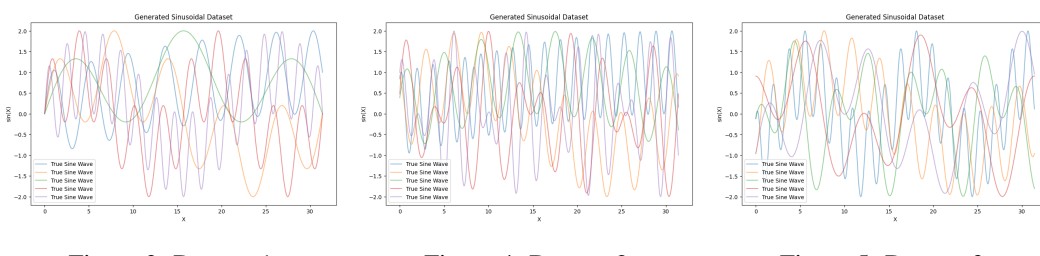

Figure 3: Dataset 1         Figure 4: Dataset 2         Figure 5: Dataset 3

maintaining the U-shape's inherent multi-scale representation. The ongoing evolution of U-shape architectures highlights their adaptability and strength in addressing a broad range of time series forecasting challenges.

## 4 EXPERIMENTS AND RESULTS

In this section, we conduct experiments to demonstrate the efficacy of weighting techniques on different series forecasting datasets. These experiments utilize the Kernel U-Net architecture and are composed of Linear and MLP kernels. We compare Stochastic Gradient Descend (SGD), SGD with momentum (SGDM), and Adaptive SGD with momentum (Adam) with the proposed Exponentially Weighted SGDM (EW-SGDM). This comparative analysis provides a comprehensive view of how different gradient descent strategies influence the U-shape neural network architectures.

### 4.1 DATASETS

We conducted experiments with 3 synthetical datasets, composed of 5 different sin functions. Dataset 1 contains a composition of sin functions with different frequencies, Dataset 2 composes sin functions with shifts, and Dataset 3 contains sin functions with more complex patterns. Figure 3 4 5. Here, we followed the experiment setting in (Zeng et al., 2023) and partitioned the data into $[0.7, 0.1, 0.2]$ for training, validation, and testing.

### 4.2 EXPERIMENT SETTINGS.

We set the look-back window $L = 512$ and forecasting horizon $T = 512$ in experiments. The list of multiples for kernel-u-net is respectively [8,8] and the segment-unit input length is 8. The hidden dimensions are 128 all layers. The input dimension is 1 as we follow the channel-independent setting in (Zeng et al., 2023) and (Nie et al., 2023). The learning rate is selected in [0.00001, 0.00005, 0.0001] for Adam and [0.001, 0.005, 0.01] for SGD methods. We examined several different weights $W \in \{4, 6, 8\}$. The momentum is set to be 0.9 for all configurations. The training epoch is 50 and the patience is 20 in general. Following previous works (Wu et al., 2021), we use Mean Squared Error (MSE) as the core metrics to compare performance for Forecasting problems.

### 4.3 RESULTS

As shown in Figure 6, the EW-SGDM method amplifies the gradient updates for higher-level parameters, and avoids small gradients at high-level layer parameters, leading to faster convergence in training. In addition, Figures 7, 8, and 9 demonstrate that EW-SGDM achieves quicker convergence compared to standard SGDM and comparable convergence curve to Adam on the training set. Furthermore, Figures 13, 14, and 15 demonstrate the EW-SGDM method outperforms both SGDM and Adam optimizers on the test set in terms of mean squared error (MSE), reaching a lower MSE across multiple experiments, demonstrating better generalization.

## 5 CONCLUSION

In conclusion, we have introduced an algorithm called exponentially weighted stochastic gradient descent (EW-SGDM), aimed at addressing the challenge of time complexity in training deep models

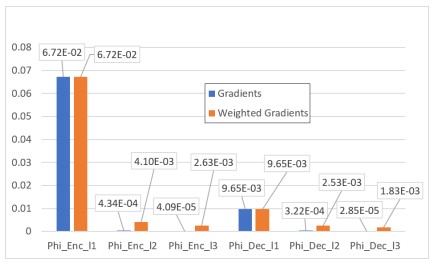

Figure 6: The exponential weight increases the absolute value of gradients for parameters in each levels in Encoder and Decoder.

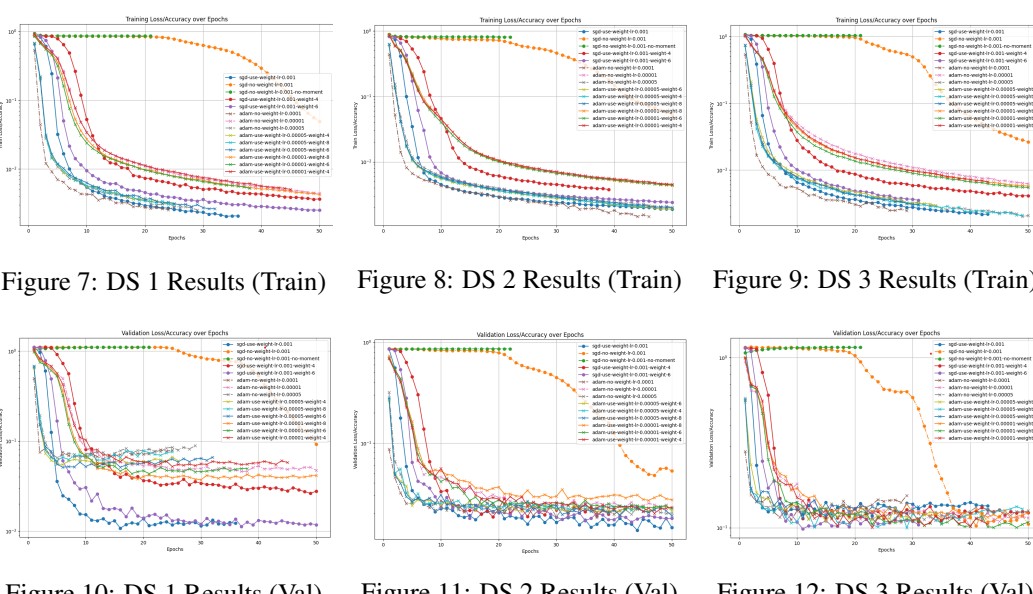

Figure 7: DS 1 Results (Train)   Figure 8: DS 2 Results (Train)   Figure 9: DS 3 Results (Train)

Figure 10: DS 1 Results (Val)   Figure 11: DS 2 Results (Val)   Figure 12: DS 3 Results (Val)

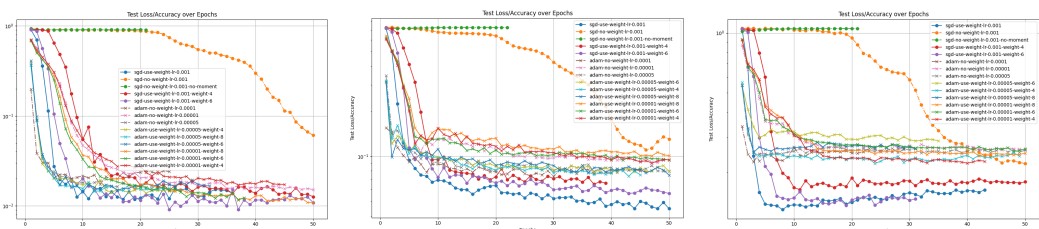

Figure 13: DS 1 Results (Test)   Figure 14: DS 2 Results (Test)   Figure 15: DS 3 Results (Test)

for time series forecasting. Our analysis revealed that current methods, despite ensuring linear time complexity, suffer from significant delays in learning high-level features. EW-SGDM offers a solution by achieving constant time complexity, as demonstrated both theoretically and empirically. Through extensive evaluations on synthetic datasets, we demonstrated that our method not only reduces computational complexity but also enhances model generalization. The future work may includes adapting this algorithms to applications such as image processing or text generations.

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
