# OpenReview forum: "Learning K-U-Net in Constant Complexity with Application to Time Series Forecasting"
_ICLR.cc/2025/Conference — ICLR 2025 Conference Withdrawn Submission_

### Official Review · Reviewer_7YYG · 2024-11-03

**Soundness:** 1
**Presentation:** 1
**Contribution:** 2
**Rating:** 3
**Confidence:** 3

**Summary:**

This paper considers time-series forecasting problem that forecasts $x_{L+1},\dots,x_{L+T}$ using $x_{1},\dots x_{L}$, where $x\in\mathbb{R}^M$ is a feature vector.
They introduce a technique they call exponentially weighted SGD with moment (EW-SGDM) to make the "learning complexity" of a time-series forecasting model, Kernel U-Net, constant to the output sequence length $T$.
This method heuristically sets learning rates for each layer to encourage learning of deeper layers.
They conducted experiments on simple synthetic datasets.

**Strengths:**

- The proposed method seems to be simple and usable for various tasks.

**Weaknesses:**

- The mathematical notations are not appropriately used and thus hard to follow. For example, although $W^{(l)}$ is a weight, and thus, expected to be a matrix and $S$ is a hyperparameter, and thus, expected to be a scalar, Eq (2) says $W^{(l)}=S^{l-1}$.
- The reason that the proposed method reduces complexity is not explained. The description, which should be around L268, is missing.
- Experiments are limited to simple synthetic experiments using sine curves.

**Questions:**

It would be appreciated if the legends of Figures 7-15 were described in a more human-friendly manner.

---

### Official Review · Reviewer_ufVf · 2024-11-03

**Soundness:** 1
**Presentation:** 1
**Contribution:** 1
**Rating:** 1
**Confidence:** 4

**Summary:**

The paper introduces an algorithm called Exponentially Weighted Stochastic Gradient Descent with Momentum (EW-SGDM), aimed at reducing the training complexity of deep learning models on time series forecasting tasks. The approach is tested on the Kernel U-Net, a U-Net variant for time series forecasting. The core contribution lies in the identification of significant redundancy in the low-level feature learning within U-Net models. By strategically ignoring these redundancies, EW-SGDM optimizes computational resources. The paper presents experiments on multiple synthetic time series datasets.

**Strengths:**

- Originality: the paper observes significant redundancy in patch-based training of time series models, which can lead to faster training behavior. Unfortunately, the rest of the paper fails to sufficiently build on this insight to present a contribution of wide interest to the ICLR community.

**Weaknesses:**

It is not clear what this paper tries to contribute:
- If an optimization paper, it should carefully compare the proposed optimization method against standard and SOTA optimization methods, on a variety of settings. In particular, it should include a comparison against the Layer-wise Adaptive Rate Scaling (LARS, cf. https://arxiv.org/abs/1708.03888 )
- If a time series paper, it should compare the Kernel U-Net against other time series forecasting methods.
- In both cases, the experimental validation is severely lacking, and the paper only provides results on synthetic datasets. There should be a comprehensive experimental validation, including real-world datasets.
- There should comprehensive analysis of the learning dynamics, with and without the proposed method, to assess whether the proposed training algorithm indeed reaches its goals.

**Questions:**

- Line 097: as written, the lookback window sees into the future from $t$, shouldn't it be defined as $(x_{t-L+1}, \ldots, x_t)$ ?
- Line 124: a `\ref` is defined without a target.
- Line 159: it's not clear what is intended by "the patch length" (S); please define explicitly.
- Line 160: should reference Figure 2, not Figure 1. Besides, the real Figure 1 is not introduced or explained in the text.
- Line 184-188: the claims here are provided without citation or analysis.
- Line 446: the reference should be fixed (authors separated by commas in BibTeX).

---

### Official Review · Reviewer_rxF1 · 2024-11-04

**Soundness:** 2
**Presentation:** 2
**Contribution:** 2
**Rating:** 3
**Confidence:** 3

**Summary:**

The paper introduces a novel exponentially weighted stochastic gradient descent algorithm - proposed to improve the training speeds of Kernel U-Nets. The method is tested on a synthetic dataset for multivariate/multihorizon forecasting, showing faster training convergence and OOS performance improvements.

**Strengths:**

The introduction of transformer based models has significantly increased the sizes of SOTA models for time series forecasting. While inference speed gets more focus, improvements in training efficiency can help to reduce the computational requirements of large models - democratising research and reducing the environmental impact of their training.

**Weaknesses:**

However, there are several key limitations in the proposed method, that raise some questions:

1. *Experimental limitations*: While applications of Kernel U-Nets have been studied, it is not immediately clear that they are the state-of-the-art for multivariate/multihorizon forecasting - with many other more competitive models having been proposed in Table 1 as well. As such, detailed performance comparisons against other SOTA and baseline models for multivariate/multihorizon forecasting are required, using non-synthetic real-world datasets. Actual wall clock time of the training methods would also be beneficial to see how much faster the model trains on the same hardware, assuming performance is similar to baselines.

2. *Generalisation of findings* - many of the motivations the paper are stated without evidence/references, and it is not immediately clear how applicable they are to the wide universe of time series forecasting problems, datasets, and architectures. E.g. that slower training convergence is a result of gradient imbalances across layers, and that increasing gradient updates in deeper layers can lead to training improvements in general.

**Questions:**

1. How well do kernel U-Nets perform with respect to other SOTA models for multivariate multihorizon forecasting, and how much quicker do they take to train?
2. Assuming that removing overlapping patches is desirable for performance, how does the method compare to simply downsampling the data and constructing minibatches with no overlaps in batched sequences?
4. How well does the proposed approach perform on real world time series datasets?

---

### Official Review · Reviewer_HMf2 · 2024-11-05

**Soundness:** 2
**Presentation:** 1
**Contribution:** 2
**Rating:** 3
**Confidence:** 3

**Summary:**

The authors propose a constant complexity method for time series forecasting. The motivation for forecasting is not so clear though.

**Strengths:**

- Clear time series problem definition in the preliminaries
- Interesting method to use modfiy the gradient updates and study the optimization method in U-Nets

**Weaknesses:**

- The abstract is very short and vague. Not all time series methods have linear complexity.
- Need spaces between words and references
- U-Net architecture is typically not used in time series forecasting
- Some mathematical variables are not italicized
- State-of-the-art time series baselines are missing from Table 1, e.g., DeepAR, MQ-CNN, MQTransformer, ETS and foundation models, e..g, Chronos (Ansari et. al, "Chronos: Learning the language of time series", 2024)
- There are references missing in Section 2.2
- The complexities of current methods in Section 2.2 can be move to an appendix
- This paper seems very preliminary and in its current state is not ready for publication.

**Questions:**

1. How tied is the method to a U-Net architecture?
2. What motivates the authors to use a U-Net?

---

### Note · Authors · 2024-11-22

I have read and agree with the venue's withdrawal policy on behalf of myself and my co-authors.